# Identification of Pathogenic and Opportunistic Yeasts in Pigeon Excreta by MALDI-TOF Mass Spectrometry and Their Prevalence in Chon Buri Province, Thailand

**DOI:** 10.3390/ijerph20043191

**Published:** 2023-02-11

**Authors:** Rungnapa Nualmalang, Natthapaninee Thanomsridetchai, Yothin Teethaisong, Passanesh Sukphopetch, Marut Tangwattanachuleeporn

**Affiliations:** 1Department of Medical Sciences, Faculty of Allied Health Sciences, Burapha University, Mueang, Chonburi 20131, Thailand; 2Department of Medical Technology, Faculty of Allied Health Sciences, Burapha University, Mueang, Chonburi 20131, Thailand; 3Research Unit for Sensor Innovation (RUSI), Burapha University, Mueang, Chonburi 20131, Thailand; 4Department of Microbiology and Immunology, Faculty of Tropical Medicine, Mahidol University, Bangkok 10400, Thailand

**Keywords:** MALDI-TOF MS, pigeon dropping, yeasts, prevalence, Thailand

## Abstract

Pigeon excreta can cause environmental and public health issues, particularly in urban and public areas. They are reservoirs of several human pathogens including fungi, bacteria, and viruses. Epidemiological data of pathogenic and opportunistic yeasts in pigeon droppings in Chon Buri, one of the most reputable tourist cities of Thailand, are scarce. The present study aimed to identify yeasts in pigeon droppings by MALDI-TOF mass spectrometry, and to study their prevalence in Chon Buri, Thailand. A total of 200 pigeon fecal samples were collected randomly from all 11 districts of Chon Buri. A sum of 393 yeast-like colonies were isolated on Sabourand’s dextrose agar and CHROMagar media. These isolates were further confirmed for their species by MALDI-TOF MS. Twenty-four yeast species belonging to 11 different genera were identified in pigeon fecal samples. *Candida* spp., predominantly *C. krusei* (14.32%), were the most prevalent yeast species. Other yeast species, including *C. glabrata* (12.73%), *C. metapsilosis* (11.93%), *Lodderomyces elongisporus* (10.87%), *C. tropicalis* (7.16%)*, C. albicans* (5.83%), and *Cryptococcus neoformans* (4.77%) were identified. This study provides valuable epidemiological data and diversity of yeasts in pigeon droppings in Chon Buri, Thailand, and also supports the use of MALDI-TOF MS for yeast identification and epidemiological surveillance.

## 1. Introduction

Pigeons (*Columba livia*) can be found pervasively in urban and public areas, especially in certain places that allow people to feed them. Since pigeons live close to people, they could harm people’s health by causing several transmissible and life-threatening infections. Pigeon droppings are becoming a significant environmental issue, along with the increasing population of pigeons. Further, pigeon excreta are among serious public health concerns because they are considered reservoirs and carriers of opportunistic and pathogenic microorganisms, such as fungi, bacteria, and viruses [1,2]. Numerous previous reports emphasized the extent of bacterial colonization in pigeon excreta. Although fungi could be one of the ethologic agents of deadly infections in humans, very few studies have focused on the colonization of opportunistic and pathogenic fungi in pigeon feces. Disseminations of fungi in pigeon droppings are typically transmitted to animals and humans through aerosol from dispersed dried excreta [3]. Among the epidemiological data of yeast colonization in pigeon feces, *Cryptococcus* spp. are the most common studies. In addition, other genera, such as *Candida* spp., *Rhodotorula* spp., and *Trichosporon* spp., can also cause life-threatening fungal infections, particularly in vulnerable and immunocompromised people [3,4].

The prevalence of pigeon droppings-dwelling yeasts is diverse depending on the geographical area. For instance, in Seoul, Korea, *Candida glabrata* (34.1%) was found to be the most prevalent yeast species presented in pigeon droppings compared to *Cryptococcus albidus* (14·3%), *Candida famata* (12.7%), and *Cryptococcus laurentii* (7·9%), respectively [5]. In the city of Moscow, *Candida albicans* was found in high abundance and frequency, as well as other opportunistic yeasts, including *Diutina catenulate, Millerozyma farinosa*, *Pichia kudriavzevii*, and *Trichosporon asahii* [6]. In western Saudi Arabia, of a total of 46 yeast-like isolates molecularly identified in pigeon droppings, *Cryptococcus neoformans* was the most prevalent yeast species (11 isolates), followed by *Cryptococcus albidus* (5 isolates), *Saccharomyces cerevisiae* (5 isolates), and *Rhodotorula mucilaginosa* (5 isolates) [1]. In Thailand, most studies have focused on the isolation and identification of *Cryptococcus* spp., especially *C. neoformans* [7,8]. Several factors could affect the diversity of yeast populations in pigeon droppings, including climate, temperature, and humidity [9].

Conventional methods for yeast identification rely on labor-intensive and time-consuming culture-dependent techniques and biochemical characterization. Traditional methods for yeast identification in clinical settings are based on phenotypic characteristics and morphologic features identified under a microscope after Gram-smear [10]. The methods for the isolation of pathogenic and opportunistic yeasts in polymicrobial specimens are typically carried out using selective media, such as Sabouraud dextrose agar and malt extract agar. However, these techniques are limited in terms of differentiating between yeast species [11]. Differential chromogenic agar, commonly CHROMagar, is a practical technique that has been used to isolate, discriminate, and identify highly prevalent yeast from clinical samples, especially *Candida* spp., by generating unique and species-specific color on CHROMagar [12]. CHROMagar is a simple and effective method for yeast discrimination from polymicrobes, but has poor specificity (62%) for bronchial secretion samples [12]. Recently, in the genomic era, molecular techniques such as polymerase chain reaction (PCR) and sequencing have become more popular for fungal identification, taxonomy, and epidemiological surveillance. Molecular techniques are known to have higher sensitivity and specificity, and are more rapid compared to a culture-based method. In several current studies, fungal identification using molecular approaches is targeted at the internal transcribed spacer (ITS) regions of the fungal genome [13].

In addition to culture and molecular identification, matrix-assisted laser desorption/ionization time-of-flight mass spectrometry (MALDI-TOF MS) has emerged as a robust and reproducible tool for microbial identification and diagnosis, including yeasts. Currently, it has increasingly been used in several microbiological laboratories globally for not only microbial identification, but also for strain typing, epidemiological studies, and antimicrobial resistance profiling. The basic principle of MALDI-TOF MS for microbial identification relies on peptide-mass fingerprinting (PMF). The interpretation of unknown microbes is achieved by comparing their PMF in the database or by matching the MS spectra of biomarkers of the unknown microbe with the proteome database [14]. Due to MALDI-TOF MS being able to detect a variety of pathogens rapidly, it has extensively been employed to identify microorganisms, such as bacteria, yeasts and molds, from clinical samples routinely [15]. Several studies have reported that this technique is rapid and exhibits high accuracy in fungal identification from clinical and environmental specimens [16,17,18]. It has also been implemented for biological, clinical, and biotechnological research. Moreover, MALDI-TOF can be used as a useful and effective technique for antifungal susceptibility profiling [19]. In this regard, the present study aimed to utilize MALDI-TOF MS for yeast identification in pigeon feces and to study the occurrence and diversity in Chon Buri, Thailand. Being situated on the eastern coastal part of Thailand, near to the main airport, Chon Buri attracts millions of domestic and international tourists each year. The most famous area is Pattaya. Chon Buri province is one of the most popular destinations for tourists. There are approximately 1,500,000 people living in 11 districts in a total area of 4363 km^2^ (12°30′ N–13°43′ N, 100°45′ E–101°45′ E).

## 2. Materials and Methods

### 2.1. Study Area and Collection of Pigeon Excreta

The present study involved the random collection of a total of 200 samples of pigeon excreta from different locations in 11 districts of Chon Buri province, Thailand, specifically from public areas (e.g., public parks) in the urban areas of each district that permit people to feed the pigeons, between March and June in 2021, as presented in Table 1. The specimens were collected using a sterile container and were subsequently processed within 6 h of collection. The predominant features of pigeons are that they always live together as a flock in the same place. They live in public areas close to humans, while other birds are not as human friendly and do not live together with pigeons. Therefore, we can easily discriminate pigeon feces from that of other birds. The characteristics of pigeon excreta are illustrated in Appendix A. They are usually larger in size than other birds’ excreta. Pigeon droppings can be white, gray, black, or dark green in color, depending on what they ate.

### 2.2. Yeast Isolation and Culture

One gram of dried and/or fresh pigeon feces sample was diluted in saline solution (0.85% NaCl) in a ratio of 1:9. The diluted samples were mixed thoroughly for 5 min and a 100 µL aliquot of supernatant was spread evenly onto Sabouraud dextrose agar (SDA) (Himedia™, Mumbai, India) supplemented with 0.4 g/l chloramphenicol [3]. Each plate was incubated at 37 °C for 3 consecutive days. After incubation, colonies showing yeast-like phenotypes were selected for further phenotypical characterization on CHROMagar candida and species identification and confirmation by a MALDI-TOF MS technique.

### 2.3. Yeast Identification by MALDI-TOF Mass Spectrometry

For MALDI-TOF MS analysis, the yeast-like isolates were prepared using protein extraction methods. Briefly, yeast colonies were taken using a 1 µL inoculating loop and transferred into 300 µL of HPLC-grade water followed by mixing thoroughly until all components were completely homogeneous. Then, 900 µL absolute EtOH was added and mixed thoroughly. The microbial materials were centrifuged at 13,000–15,000 rpm for 2 min. The supernatant was carefully discarded without disturbing the pellet. Following the repeat of aforementioned step, the resultant pellet was air-dried for at least 5 min at room temperature before adding 25 µL of 70% aqueous formic acid. The solution was mixed by pipetting the solution up and down until the pellet was completely suspended. Twenty-five microliters of 100% acetonitrile was subsequently added and mixed by pipetting 2–3 times. The solution was centrifuged at 13,000–15,000 rpm for 2 min followed by loading 1 µL of the supernatant onto a vacant sample position of the MALDI target plate. Once the plate was air-dried at room temperature, 1 µL of HCCA matrix solution was added to overlay each sample and allowed to air-dry at room temperature again. The fungal species identification was achieved using MALDI Biotyper^®^ (MBT) software (Bruker, Germany) based on MALDI-TOF MS (Bruker, Germany).

## 3. Results

Of the 200 pigeon dropping samples collected from various locations in Chon Buri province, a total of 393 yeast-like isolates were identified on SDA and CHROMagar medium, as summarized in Table 1. The fecal samples were mostly collected in Muang Chon Buri (61 samples), the capital city of Chon Buri province. The colony morphology and characteristics of the yeast-like isolates recovered from pigeon feces on SDA supplemented with 0.4 g/L chloramphenicol and on CHROMagar are illustrated in Appendix A. Different yeast species exhibited distinct phenotypes on CHROMagar medium (Appendix A), with exceptions for some yeast species including *Arthographis kalrae*, *Candida carpophila*, *Candida guillermontii*, *Candida kefyr*, *Candida lusitaniae*, *Candida metapsilosis*, *Candida nivariensis*, *Candida nivariensis*, *Candida orthopsilosis*, *Candida rugosa*, *Cryptococcus neoformans*, *Cyberlindnera fabianii*, *Cyberlindnera misumaiensis*, *Kazachstania telluris*, *Ogataea polymorpha*, *Pichia manshurica*, and *Saccharomyces cerevisiae*.

Distinct yeast species, such as *Candida krusei*, *Candida glabrata*, *Candida metapsilosis*, *Lodderomyces elongisporus*, *Candida tropicalris*, and *Trichosporon asahii*, have different main spectra profiles of MALDI-TOF MS, allowing the MALDI-TOF MS approach to identify and discriminate between different yeast species (Figure 1). Dendrogram analysis confirmed that the species were well distinguished by MALDI-TOF MS (Figure 2).

The results of the yeast identification carried out by MALDI-TOF MS are summarized in Figure 3. MALDI-TOF MS identified a total of 24 different yeast species belonging to 11 different genera including *Candida*, *Lodderomyces*, *Trichosporon*, *Cryptococcus*, *Kazachstania*, *Saccharomyces*, *Cyberlindnera*, *Ogataea*, *Arthographis*, *Debaryomyces*, and *Pichia*. *Candida* spp. were found to be the most prevalent (274 isolates, 69.72%), including *C. krusei* (14.32%), *C. glabrata* (12.73%), *C. metapsilosis* (11.93%), *C. tropicalis* (7.16%), *C. albicans* (5.83%), and other species (78 isolates, 19.84%). *L. elongisporus*, *T. asahii*, *C. neoformans*, *K. telluris*, and other species had occurrences of 10.87, 6.36,4.77, 4.77, and 4.83%, respectively. Mueang Chon Buri district showed the highest abundance of yeasts colonized in pigeon droppings, followed by Bo Thong, Phan Thong, and Bang Lamung. *Candida* spp. were the most commonly found yeasts in pigeon droppings in all districts, except for Koh Sichang, where *Kazachstania telluris* had a higher prevalence compared to *Candida* spp. (Figure 3). Interestingly, *L. elongisporus* was found in the highest abundance and frequency in Mueang Chon Buri district compared to the other districts (Figure 3).

In a comparison between CHROMagar and MALDI-TOF MS, among 393 yeast-like isolates, this study found that CHROMagar identified 147 isolates (37.4%) differently to MALDI-TOF MS, as summarized in Table 2. The common yeast species that were found to be different by CHROMagar included *Candida tropicalis*, *Candida albicans*, *Candida krusei*, *Candida parapsilosis*, and *Lodderomyces elongisporus*. The following yeast species were not identified by CHROMagar, but were identified by MALDI-TOF MS: *Arthographis kalrae*, *Candida capophila*, *Candida guillermontii*, *Candida kefyr*, *Candida lusitaniae*, *Candida metapsilosis*, *Candida nivariensis*, *Candida nivariensis*, *Candida othopsilosis*, *Candida rugosa*, *Cryptococcus neoformans*, *Cyberlindnera fabianii*, *Cyberlindnera misumaiensis*, *Kazachstania telluris*, *Ogataea polymorpha*, *Pichia manshurica*, and *Saccharomyces cerevisiae*.

## 4. Discussion

Pigeon excreta are known as a potential source and carrier of pathogenic and opportunistic yeasts that can contaminate the environment and be transmitted to animals and humans. Pigeons have a relatively high body temperature (approximately 40 °C), and yeasts cannot grow well in this temperature. However, their dried excreta comprise low molecular weight nitrogenous compounds that are suitable conditions to promote fungal growth [20]. Several yeasts prefer to grow in organic compound-rich soil similar to pigeon excreta. This is a good growth-promoting factor for yeasts. Since pigeons live in public areas, their excreta are of great environmental and public health concern [21]. The transmission of yeast to humans is mainly mediated by aerosolization from dried excrement [20]. The symptoms of yeast infections range from mild to severe depending on the host’s immunological status. The mortality rate from invasive yeast infections is substantially higher in immunocompromised or immunosuppressed patients compared to healthy individuals [22].

There are several techniques that have been used to identify and diagnose fungal species, including culture- and biochemical-dependent and molecular-based techniques. In addition to these methods, MALDI-TOF MS is an emerging and interesting technique that has been increasingly used for the identification of microbial pathogens, including fungi. A previous study reported that MALDI-TOF MS had a sensitivity and specificity of up to 94.6% and 99%, respectively, for the identification of pathogenic yeasts in clinical samples [23]. Similarly, Zhao et al. (2017) reported that using the Bruker MALDI-TOF MS platforms accurately identified 90.1% of yeast species compared to 28S nrDNA and ITS sequencing analyses [10]. Lee and colleagues (2019) employed the ASTA MALDI-TOF MS system for pathogenic yeast identification from clinical specimens. The study reported that MALDI-TOF MS correctly identified 270 out of 284 samples, achieving 95.1% accuracy [24]. These findings indicate that MALDI-TOF MS is a robust, rapid, and effective tool for pathogenic and opportunistic yeast identification from both clinical and environmental samples. Although CHROMagar has some limitations in terms of yeast identification, it is among the methods most commonly used in microbiological laboratories, particularly in resource-constrained areas, for phenotypic characterization and differentiation of *Candida* spp. and other yeast species. In the present study, CHROMagar provided valuable data, especially for *Candida* spp., which is the most prevalent yeast species colonized in pigeon droppings in Chon Buri, Thailand. Nevertheless, the sensitivity and specificity of CHROMagar for the identification and discrimination of *Candida* spp. were 68 and 82%, respectively [12]. However, with CHROMagar being of lower sensitivity and specificity than MALDI-TOF MS, this study provides strong evidence to support the use of MALDI-TOF MS for not only yeast species identification, but also for the studies of yeast diversity and epidemiological surveillance in pigeon excreta. Moreover, MALDI-TOF MS has been employed to study antifungal susceptibility profiling in *Candida* spp. [19]. Hence, future work could focus on the incidence of antifungal resistance in pigeon excreta using MALDI TOF MS. However, MALDI TOF MS has some limitations, including high maintenance costs, the need for skilled personnel, and the fact that yeast cultured on different media may result in changes in the protein expression profile [10]. Furthermore, it is well established that CHROMagar medium is unable to identify or distinguish between the following yeast species: *Arthographis kalrae*, *Candida carpophila*, *Candida guillermontii*, *Candida kefyr, Candida lusitaniae*, *Candida metapsilosis*, *Candida nivariensis*, *Candida nivariensis*, *Candida orthopsilosis*, *Candida rugosa*, *Cryptococcus neoformans*, *Cyberlindnera fabianii*, *Cyberlindnera misumaiensis*, *Kazachstania telluris*, *Ogataea polymorpha*, *Pichia manshurica*, and *Saccharomyces cerevisiae* [25,26]. Hence, this limitation restricts the use of CHROMagar to study the diversity and epidemiological surveillance of yeasts in environmental samples. In this regard, we employed the MALDI-TOF MS technique to identify yeast species colonized in pigeon excreta together with studying the prevalence and diversity in Chon Buri province, Thailand.

Most of the previous studies on yeast populations in pigeon droppings focused on *Cryptococcus* spp., in particular *C. neoformans*, since they can cause invasive life-threatening cryptococcosis, which is an infection that affects the lungs, brain, and spinal cord in humans, predominantly in immunocompromised individuals. The prevalence of *C. neoformans* in pigeon droppings varies by geographical area and weather conditions. *C. neoformans* can be found ubiquitously in the environment and in animals [8,27,28]. Among more than 30 species in the genus *Cryptococcus*, the *C. neoformans*/*C. gattii* species complex is the most common yeast pathogenic to humans. This species can be frequently found in bird droppings and in the environment surrounding the trees including leaves, flowers, decaying wood, and soil. The transmission takes place through the inhalation of airborne spores, eventually causing life-threatening infection [29]. Within the environment, this species gains selective advantage by adapting to new habitats and infecting new hosts. Microorganisms live in a range of environmental conditions and can tolerate physical stress and nutrient and energy starvation [30].

The present study found that *Candida* spp. was the most prevalent yeast identified in the pigeon droppings in almost all regions of Chon Buri. This finding is consistent with a study conducted by Magalhaes Pinto et al. (2019) that found that 41.8% of the yeasts identified in pigeon extreta in Brazil were *Candida* yeasts (*C. parapsilosis* complex, *C. tropicalis*, *C. krusei*, *C. glabrata,* and *C. rugosa*). Furthermore, these isolated yeasts had several virulence factors, including the ability to adhere to human bronchial epithelial cells (HBEC), biofilm formation, and the secretion of lytic enzymes, such as hemolysins, proteinases, and phospholipases [3]. In addition, our result is also in agreement with a study in Seoul, Korea, where it was reported that *Candida* spp. was the most predominant yeast found in pigeon feces [5]. There are approximately 150 species in the genus *Candida.* Most of them are non-pathogenic and are considered endosymbionts of humans; however, they can cause severe infections in vulnerable humans, especially patients with immunodeficiency. Resistance to antifungal agents of *Candida* spp. is of great concern. A study showed that *Candida* species may express resistance to azoles [3]. As much as 80% of infections are caused by *C. albicans*, with the remainder caused by *C. glabrata*, *C. tropicalis*, *C. krusei*, and *C. dubliniensis* [31]. *C. albicans* can be both commensal flora and pathogenic to humans in both healthy and immunocompromised people by causing candidiasis. The disease may appear superficially (e.g., oral, vaginal, mucocutaneous) or profoundly (e.g., myocarditis, septicemia). In women, 75% will suffer with vaginal candidiasis at least once during their lifetime [32].

Regarding *L. elongisporus* in the pigeon droppings isolated in this study, we found a high frequency of approximately 11% (41 isolates), which is higher than a study conducted in western Saudi Arabia, where the incidence of *L. elongisporus* in pigeon feces was approximately 4.34% [1]. *L. elongisporus* is an opportunistic infection of great concern in immunocompromised patients. In Thailand, *L. elongisporus* can be found in sugarcane leaves and palm shells. The number of infections with *L. elongisporus* is underestimated. It is often misidentified as *Candida parapsilosis* by the VITEK 2 yeast identification system. The pathogenesis of *L. elongisporus* is still unknown; however, a progressive fungemia has been reported in a 71-year-old woman [33], and recently in a premature, extremely low-birth-weight neonate [34]. In addition, the first case of meningitis caused by *L. elongisporus* was reported a 62-year-old man [35].

*Trichosporon* species are characteristically yeast-like organisms that can be found abundantly in nature. *Trichosporon* colonizes as commensal flora of the human skin, vagina, and gastrointestinal tract. Recently, *Trichosporon* spp., predominantly *Trichosporon asahii*, has increasingly been recognized as an opportunistic yeast that causes infections in immunocompromised individuals and rarely in immuno-competent patients. Infections caused by *Trichosporon* spp. can be classified as superficial infections and invasive infections [36]. Previous studies reported that *T. asahii* is rare in the clinical setting, but often found in pigeon excreta [37]. In this study, we found that *T. asahii* accounted for 13% of the species in pigeon excreta, supporting a previous study by Pakshir et al. (2019) that isolated this yeast from pigeon excreta in southern Iran [38].

*Kazachstania* spp. are ubiquitous in nature and are classified in the Saccharomycetaceae family. Although they are considered a rare fungal pathogen, 13 cases of infections caused by *Kazachstania* spp. Were reported in Strasbourg University Hospital, Strasbourg, France. *K. telluris* is an opportunistic yeast that can cause fungemia in humans. This particular yeast is frequently confected with *C. albicans* [39]. In the present study, we found *K. telluris* colonized in the pigeon excreta at a rate of approximately 4.77%. Taken together, the findings from the present study indicate that a variety of pathogenic and opportunistic yeasts colonize in pigeon droppings. Hence, care should be taken to reduce human exposure to pigeon excreta.

## 5. Conclusions

Pigeon droppings collected in Chon Buri province, Thailand, present diverse yeast species, predominantly *Candida* spp., that are highly prevalent in several regions of Chon Buri. Among the identified yeasts, *Cryptococcus neoformans* and *Candida* spp. are particularly known to cause life-threatening cryptococcosis and candidiasis, respectively. MALDI-TOF MS is a rapid, easy-to-use, and robust tool for yeast identification. It is a promising alternative technique to CHROMagar for yeast identification, with higher performance and accuracy. This study provides important epidemiological data regarding the diversity and prevalence of pathogenic and opportunistic yeasts that could have the potential to cause infections, especially in vulnerable and immunosuppressed individuals.

## Figures and Tables

**Figure 1 ijerph-20-03191-f001:**
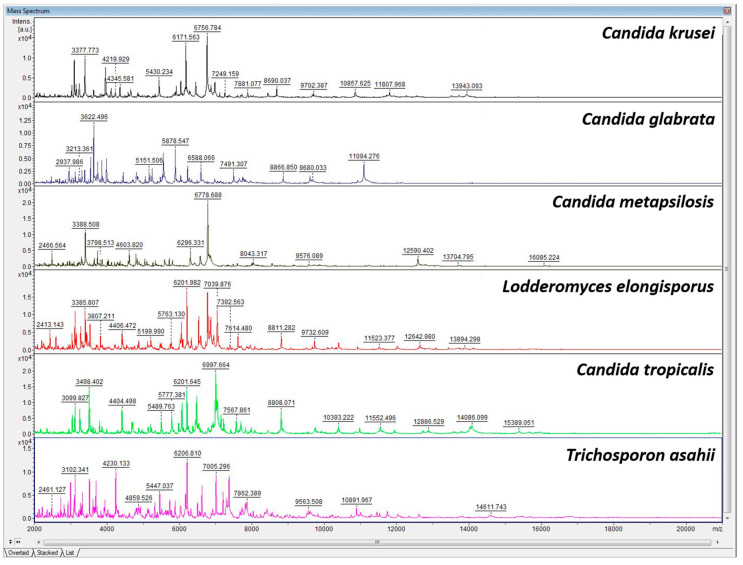
Representative spectra profiles of six common yeast species colonized in pigeon droppings.

**Figure 2 ijerph-20-03191-f002:**
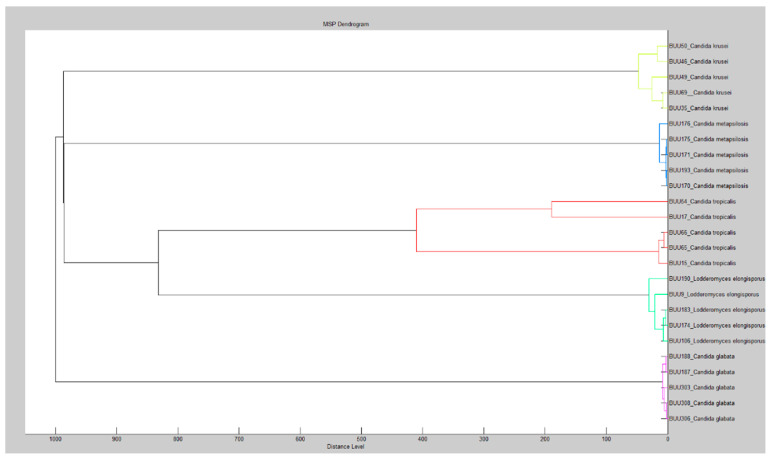
Dendrogram from MALDI-TOF mass spectrometric analysis.

**Figure 3 ijerph-20-03191-f003:**
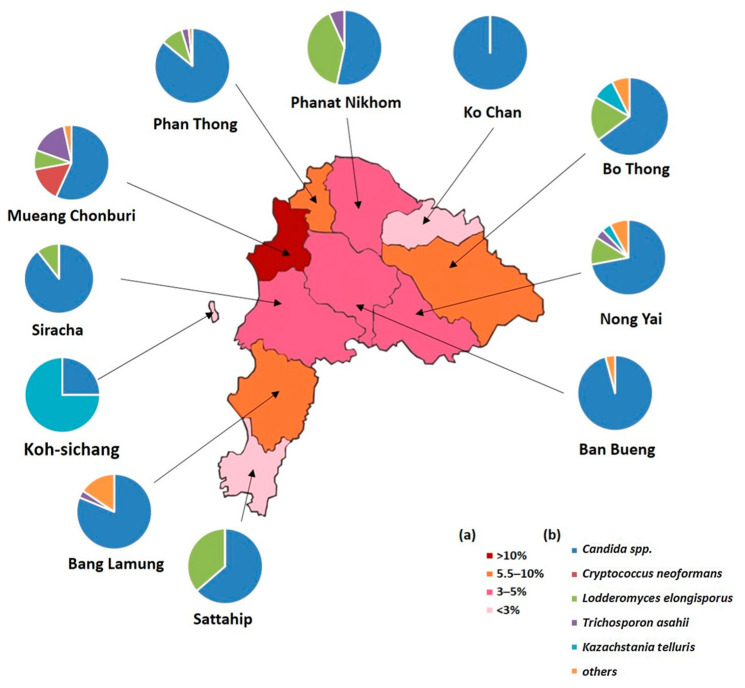
Map of Chon Buri province, Thailand: (**a**) percentage of pigeon excreta containing yeasts; (**b**) pie charts indicating the prevalence of yeast species found in pigeon excreta samples.

**Table 1 ijerph-20-03191-t001:** Pigeon excreta sample collection.

Location	No. of Sample Collected	No. of Colonies Observed	No. of Yeast Species
Ban Bueng	21	24	5
Bang Lamung	19	32	12
Bo Thong	13	54	10
Ko Chan	10	15	2
Koh Sichang	6	16	4
Mueang Chonburi	61	118	14
Nong Yai	12	25	11
Phanat Nikhom	13	15	6
Phan Thong	17	64	13
Sattahip	13	11	3
Si Racha	15	19	7

**Table 2 ijerph-20-03191-t002:** Summary of 147 yeast isolates differently identified by CHROMagar and by MALDI-TOF MS.

CHROMagar Characteristics	Yeast Species Identified by CHROMagar	Yeast Species Identified by MALDI-TOF MS	No.
blue	*Candida tropicalis*	*Candida metapsilosis*	2
cream-white	*Candida tropicalis*	*Candida orthopsilosis*	1
dark blue	*Candida tropicalis*	*Candida nivariensis*	1
green	*Candida albicans*	*Cyberlindnera misumaiensis*	1
green	*Candida albicans*	*Trichosporon asahii*	1
light blue	*Candida tropicalis*	*Candida carpophila*	1
light blue	*Candida tropicalis*	*Candida metapsilosis*	44
pale pink	*Candida krusei*	*Pichia manshurica*	1
pale purple	*Candida krusei*	*Candida carpophila*	2
pale red	*Candida parapsilosis*	*Kazachstania telluris*	1
pink	*Candida parapsilosis*	*Candida carpophila*	1
pink	*Candida parapsilosis*	*Candida kefyr*	2
pink, gray, purple	*Candida glabrata*	*Candida lusitaniae*	15
purple	*Candida krusei*	*Candida carpophila*	3
purple	*Candida krusei*	*Candida nivariensis*	1
purple	*Candida krusei*	*Candida guillermontii*	1
purple	*Candida krusei*	*Cyberlindnera fabianii*	1
purple	*Candida krusei*	*Kazachstania telluris*	1
red	*Candida parapsilosis*	*Kazachstania telluris*	16
pink to purple	*Candida krusei*	*Candida guillermontii*	2
turquoise	*Lodderomyces elongisporus*	*Candida carpophila*	1
turquoise	*Lodderomyces elongisporus*	*Candida nivariensis*	1
turquoise	*Lodderomyces elongisporus*	*Candida orthopsilosis*	1
turquoise	*Lodderomyces elongisporus*	*Candida tropicalis*	1
white	*Candida parapsilosis*	*Arthographis kalrae*	2
white	*Candida parapsilosis*	*Candida carpophila*	7
white	*Candida parapsilosis*	*Candida nivariensis*	3
white	*Candida parapsilosis*	*Candida rugosa*	5
white	*Candida parapsilosis*	*Candida tropicalis*	1
white	*Candida parapsilosis*	*Cryptococcus neoformans*	19
white	*Candida parapsilosis*	*Cyberlindnera fabianii*	2
white	*Candida parapsilosis*	*Ogataea polymorpha*	3
white	*Candida parapsilosis*	*Saccharomyces cerevisiae*	4

## Data Availability

All data generated during this study are presented in an analyzed format in this manuscript. Raw datasets used for analysis in this study are available from the corresponding author on reasonable request.

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
