# Peer review of "Identification of Pathogenic and Opportunistic Yeasts in Pigeon Excreta by MALDI-TOF Mass Spectrometry and Their Prevalence in Chon Buri Province, Thailand"

_ijerph, 2023, doi:10.3390/ijerph20043191_

Round 1

Reviewer 1 Report

Diversity and prevalence of fungal species in environmental settings represent important data with direct impact on human health. Although I see value in the information presented in this manuscript, I think authors should focus on the diversity and prevalence of the different species identified, and disregard the comparison with CHROMagar, as it is well known that the latter is not an appropriate method for ID.

The manuscript needs an extensive English revision. Although the paper can be understood, I detect syntax and grammar errors, some typos and punctuation problems. Also, I see plenty spelling errors of species names.

The objective of the paper is written twice in different paragraphs within the introduction. This should be fixed and streamlined.

Pg 2, ln 82-81. The information regarding Chon Buri province should be in the introduction.

What were the characteristics of the locations where samples were taken? Urban, rural? Private, public?

What characteristics determine that the samples collected were from pigeons and not other type of bird?

What CHROMagar was used? Candida, Candida plus?

My main issue with the work presented is that CHROMagar is used as a method of phenotypical identification, and further used as a comparison against MALDI-TOF. CHROMagar is not meant to be used for identification, but for isolation of yeasts; misidentification is a given. What is the basis for the ID of, for example, C parapsilosis and L elongosporus by CHROMagar when this method does not account for the ID of said species? If authors wished to compare MALDI-TOF ID with phenotypical ID, other biochemical methods should had been used. Auxonogram would have been most appropriate.

Pg 3, ln120-124. This is not a sentence. What did the authors meant to say in this part?

Figure 1 could be taken out. I do not think this figure contributes to the manuscript.

Pg 4, ln 132-136. Why are the authors focusing on the spectra profiles and dendrogram of only these six species? Spectra profiles should be different for all species for MALDI-TOF to properly identify them. The efficiency of the method for the ID of these yeasts has been already reported. I fail to see the reason to include this Figure at all (fig 3). The paper should focus only on the diversity of species found by MALDI-TOF.

Authors must be careful when using “C.”. It may refer to Candida or Cryptococcus.

Both “Candida krusei” and “Pichia kudriavzevii” are used within the text. P kudriavzevii should be used.

Pg 8, ln 219-220. What are the references for these studies?

Author Response

Dear reviewer,

Reviewer #1

  • Diversity and prevalence of fungal species in environmental settings represent important data with direct impact on human health. Although I see value in the information presented in this manuscript, I think authors should focus on the diversity and prevalence of the different species identified, and disregard the comparison with CHROMagar, as it is well known that the latter is not an appropriate method for ID.

  Response:  We thank the reviewer for pointing this out. Yes, we do agree with you that CHROMagar may not appropriate method for ID.  Although CHROMagar has some limitations, but it still has been used in several microbiological laboratories for phenotypic characterization and differentiation of Candida spp, and other yeasts.  Here in the present study, CHROMagar provided some valuable data, particularly for Candida spp. which are the most prevent yeast species colonized in pigeon droppings in Chon Buri, Thailand. More importantly, being lower sensitivity and specificity of CHROMagar compared to MALDI-TOF MS, the study provides the strong evidence to support the MALDI-TOF MS for not only yeast species identification, but also for diversity and epidemiological surveillance studies.  

 On top of that, several previous studies used CHROMagar as a comparator for validating the performance of MALDI-TOF MS. For instance, studies reported by Lu et al (2021) and EVREN et al. (2022)

  References:

  • Evren, E., Gocmen, J. S., Istar, E. H., Yavuzdemir, S., Tekeli, A., Yavuz, Y., & Karahan, Z. C. (2022). Medically important Candida spp. identification: an era beyond traditional methods. Turk J Med Sci, 52(3), 834-840. doi:10.55730/1300-0144.5380
  • Lu, J. J., Lo, H. J., Lee, C. H., Chen, M. J., Lin, C. C., Chen, Y. Z., . . . Wang, S. H. (2021). The Use of MALDI-TOF Mass Spectrometry to Analyze Commensal Oral Yeasts in Nursing Home Residents. Microorganisms, 9(1). doi:10.3390/microorganisms9010142

  • The manuscript needs an extensive English revision. Although the paper can be understood, I detect syntax and grammar errors, some typos and punctuation problems. Also, I see plenty spelling errors of species names.

Response: Thank you for suggestion. This manuscript has been polished for English language by English expert accordingly.

  • The objective of the paper is written twice in different paragraphs within the introduction. This should be fixed and streamlined

Response:  Thank for noticing this point. We have deleted following sentence “Here in the present study, we identified and studied the prevalence of yeasts covering a variety of pathogenic and opportunistic yeasts in pigeon excreta”. So, we currently have no redundant written object in the introduction. 

  • Pg 2, ln 82-81. The information regarding Chon Buri province should be in the introduction.

Response: We have moved following sentences from method to the introduction regarding Chon Buri province; “Chon Buri province is one of the most popular destinations for tourists. There are ap-proximately 1,500,000 people live in 11 districts in a total area of 4,363 km2 (12°30′ N–13°43′ N, 100°45′ E–101°45′ E).”

  • What were the characteristics of the locations where samples were taken? Urban, rural? Private, public?

Response: The samples were collected from public areas (e.g. public parks) in urban of each district, especially the places where pigeons live as a flock and people are allowed to feed them.

  • What characteristics determine that the samples collected were from pigeons and not other type of bird?

Response:  The predominant characteristic of pigeons is they live together as a flock and always live in the same place, and they live in pubic area and close to humans. Other birds are not human friendly, and they are not live together with pigeons.  In addition, Pigeon droppings size are usually larger than others and it has white, grey, black or dark green feces depending on what they ate as shown in the picture below.

  • Pigeon dropping (b) Old world sparrows’ feces

  • What CHROMagar was used? Candida, Candida plus?

   Response:  We used CHROMagar Candida agar.

  • My main issue with the work presented is that CHROMagar is used as a method of phenotypical identification, and further used as a comparison against MALDI-TOF. CHROMagar is not meant to be used for identification, but for isolation of yeasts; misidentification is a given. What is the basis for the ID of, for example, C. parapsilosis and L. elongisporus by CHROMagar when this method does not account for the ID of said species? If authors wished to compare MALDI-TOF ID with phenotypical ID, other biochemical methods should had been used. Auxonogram would have been most appropriate.

 Response:  We thank the reviewer for this comment.  Although CHROMagar has some drawbacks for yeast identification, it has been used in serval laboratories. Some previous studies used CHROMagar for not only Candida spp. identification, but also for some other non-Candida. The example is given below

Reference:  Yas KhudhairAl-Ameri, A., & Alrufae, M. M. (2022). Phenotypic and Genotypic Identification of Yeast Species Isolated from Diabetic Foot Patients in Al-Najaf Province, Iraq. Arch Razi Inst, 77(2), 727-731. doi:10.22092/ARI.2022.357104.1975

Reference:  Lu, J. J., Lo, H. J., Lee, C. H., Chen, M. J., Lin, C. C., Chen, Y. Z., . . . Wang, S. H. (2021). The Use of MALDI-TOF Mass Spectrometry to Analyze Commensal Oral Yeasts in Nursing Home Residents. Microorganisms, 9(1). doi:10.3390/microorganisms9010142

  • Pg 3, ln120-124. This is not a sentence. What did the authors meant to say in this part?

   Response:  We thank the reviewer for pointing this out. We have change accordingly which now reads as “with exceptions for some yeast species including Arthographis kalrae, Candida carpophi-la,Candida guillermontii,Candida kefyr, Candida lusitaniae, Candida metapsillosis, Candida ni-variensis, Candida nivariensis, Candida orthopsilosis, Candida rugosa, Cryptococcus neoformans, Cyberlindnera fabianii, Cyberlindnera misumaiensis, Kazachstania telluris, Ogataea polymorpha, Pichia manshurica, and Saccharomyces cerevisiae”. 

  • Figure 1 could be taken out. I do not think this figure contributes to the manuscript.

  Response:  Thank you for your suggestion. We have moved this particular figure to the supplementary data just in case some readers may want to see yeast-like colonies on SDA and phenotypical characteristics on CHROMagar.

  • Pg 4, ln 132-136. Why are the authors focusing on the spectra profiles and dendrogram of only these six species? Spectra profiles should be different for all species for MALDI-TOF to properly identify them. The efficiency of the method for the ID of these yeasts has been already reported. I fail to see the reason to include this Figure at all (fig 3). The paper should focus only on the diversity of species found by MALDI-TOF.

Response:  We thank for pointing this out. Figure 2, which is now Figure in the revised manuscript, is the representative of 6 commonly yeast species colonized in pigeon droppings. Regarding to dendrogram, it was selected to show because we would like to highlight that MALDI-TOF MS can discriminate different yeast species which is important for the study of yeast diversity in samples recover from environment

  • Authors must be careful when using “C.”. It may refer to Candida or Cryptococcus.

         Response:   We thank for the comment. We would like to address that the nomenclature of yeast species written in this study is followed the international standard. So, we used C. followed by its specific epithet, such as C. albicans  and C. neoformans

  • Both “Candida krusei” and “Pichia kudriavzevii” are used within the text. P kudriavzevii should be used.

           Response:   We thank for the comment. We would like to address that Pichia kudriavzevii is the teleomorph of the Candida krusei.

Pg 8, ln 219-220. What are the references for these studies?

Response:  We thank for this suggestion. The following reference was added in the manuscript;

Li, H., Guo, M., Wang, C., Li, Y., Fernandez, A. M., Ferraro, T. N., . . . Chen, Y. (2020). Epidemiological study of Trichosporon asahii infections over the past 23 years. Epidemiol Infect, 148, e169. doi:10.1017/S0950268820001624

Reviewer 2 Report

This study presents epidemiological data of yeasts in pigeon  dropping in Chon Buri, Thailand with the use of MALDI-TOF MS. 16S and Internal Transcribed Spacer (ITS) ribosomal RNA (rRNA) sequencing can further strengthen the study results. 

Author Response

Dear Reviewer,

 We appreciate your valuable and constructive comments and suggestion. We would like to address that several studies have proven that MALDI-TOF is a rapid, robust and high accuracy tool, with high sensitivity (95%) and specificity (99%) for fungal identification compared to PCR and sequencing analysis. For more information can be found in the references below;

 References:

  1. Robert, M. G., Cornet, M., Hennebique, A., Rasamoelina, T., Caspar, Y., Ponderand, L., . . . Maubon, D. (2021). MALDI-TOF MS in a Medical Mycology Laboratory: On Stage and Backstage. Microorganisms, 9(6). doi:10.3390/microorganisms9061283
  2. Meena, S., Mohanty, A., Kaistha, N., Sasirekha, U., & Meena, J. (2021). Comparative Assessment of Matrix-assisted Laser Desorption Ionization-time of Flight Mass Spectrometry (MALDI-TOF-MS) and Conventional Methods in the Identification of Clinically Relevant Yeasts. Cureus, 13(6), e15607. doi:10.7759/cureus.15607
  3. Lu, J. J., Lo, H. J., Lee, C. H., Chen, M. J., Lin, C. C., Chen, Y. Z., . . . Wang, S. H. (2021). The Use of MALDI-TOF Mass Spectrometry to Analyze Commensal Oral Yeasts in Nursing Home Residents. Microorganisms, 9(1). doi:10.3390/microorganisms9010142
  4. Bonifaz, A.; Montelongo-Martínez, F.; Araiza, J.; González, G.M.; Treviño-Rangel, R.; Flores-Garduño, A.; Camacho-Cruz, A.; Tirado-Sánchez, A. [Evaluation of MALDI-TOF MS for the identification of opportunistic pathogenic yeasts of clinical samples]. Revista chilena de infectologia : organo oficial de la Sociedad Chilena de Infectologia 2019, 36, 790-793, doi:10.4067/s0716-10182019000600790.

Reviewer 3 Report

The role of pigeons and other birds as carriers of various important public health pathogens such as protozoa, fungi, bacteria, and viruses is well-known. However, this study evaluates the role of pigeon droppings as carriers of various yeasts, especially Candida species, and the usage of MALDI-TOF MS for the effective identification of these yeasts. This manuscript was well written by the authors in scientific and clear language, and it is thought the results of this study provide valuable epidemiological data for future studies conducted on the subject.

Author Response

Dear Reviewer,

      We really appreciate your positive comment and compliment on our manuscript

  Sincerely yours,

Assistant Professor Dr.  Marut Tangwattanachuleeporn

Round 2

Reviewer 1 Report

I see significant improvement in the manuscript.

Species names spelling need to be checked again (eg: "Tricosporon", "Debarymyces").

I still thing one paragraph (ln 125-129) and Figure 1 do not contribute much to the paper and could be left out.

Author Response

  • Species names spelling need to be checked again (eg: "Tricosporon", "Debarymyces").

     Response:  Thank you for pointing this out. We have corrected accordingly

  • I still thing one paragraph (ln 125-129) and Figure 1 do not contribute much to the paper and could be left out.

 Response:  Thank you for this suggestion. We do agree with your comment, but the journal requires us to add more detail to meet a total of 4,000 words. This is the reason why we keep this data in the manuscript at least it provides some data on different spectra of six commonly found yeast species in pigeon excreta